# The bZIP Transcription Factor AflRsmA Regulates Aflatoxin B_1_ Biosynthesis, Oxidative Stress Response and Sclerotium Formation in *Aspergillus flavus*

**DOI:** 10.3390/toxins12040271

**Published:** 2020-04-23

**Authors:** Xiuna Wang, Wenjie Zha, Linlin Liang, Opemipo Esther Fasoyin, Lihan Wu, Shihua Wang

**Affiliations:** Key Laboratory of Pathogenic Fungi and Mycotoxins of Fujian Province, School of Life Sciences, Fujian Agriculture and Forestry University, Fuzhou 350002, China; xiuna0304@163.com (X.W.); xxsc930901@163.com (W.Z.); lianglinlinpha@163.com (L.L.); 2018Y90100144@caac.cn (O.E.F.); lihan_wu@163.com (L.W.)

**Keywords:** *Aspergillus flavus*, bZIP transcription factor, aflatoxin B_1_, oxidative stress response, sclerotium formation

## Abstract

Fungal secondary metabolites play important roles not only in fungal ecology but also in humans living as beneficial medicine or harmful toxins. In filamentous fungi, bZIP-type transcription factors (TFs) are associated with the proteins involved in oxidative stress response and secondary metabolism. In this study, a connection between a bZIP TF and oxidative stress induction of secondary metabolism is uncovered in an opportunistic pathogen *Aspergillus flavus*, which produces carcinogenic and mutagenic aflatoxins. The bZIP transcription factor AflRsmA was identified by a homology research of *A*. *flavus* genome with the bZIP protein RsmA, involved in secondary metabolites production in *Aspergillus*
*nidulans*. The *AflrsmA* deletion strain (*ΔAflrsmA*) displayed less sensitivity to the oxidative reagents tert-Butyl hydroperoxide (tBOOH) in comparison with wild type (WT) and *AflrsmA* overexpression strain (*AflrsmA^OE^*), while *AflrsmA^OE^* strain increased sensitivity to the oxidative reagents menadione sodium bisulfite (MSB) compared to WT and *ΔAflrsmA* strains. Without oxidative treatment, aflatoxin B_1_ (AFB_1_) production of *ΔAflrsmA* strains was consistent with that of WT, but *AflrsmA^OE^* strain produced more AFB_1_ than WT; tBOOH and MSB treatment decreased AFB_1_ production of *ΔAflrsmA* compared to WT. Besides, relative to WT, *ΔAflrsmA* strain decreased sclerotia, while *AflrsmA^OE^* strain increased sclerotia. The decrease of AFB_1_ by *ΔAflrsmA* but increase of AFB_1_ by *AflrsmA^OE^* was on corn. Our results suggest that AFB_1_ biosynthesis is regulated by AflRsmA by oxidative stress pathways and provide insights into a possible function of AflRsmA in mediating AFB_1_ biosynthesis response host defense in pathogen *A*. *flavus*.

## 1. Introduction

Transcription factors (TFs) as the last link between signal flow and target genes are essential players in the signal transduction pathways. Based on the conserved DNA-binding domain, transcription factors are generally classified into structures and categories. In the Fungal Transcription Factor Database [1], 118,563 putative transcription factors found in 249 fungal and 6 oomycete strains were classified into 61 families. Among the TF families, a family of dimerizing TFs contains the conserved basic leucine zipper (bZIP) domain. The bZIP-type transcription factor as one of the largest families of dimerizing TFs is widely distributed in the all eukaryotes genomes. The first bZIP-type TF was discovered in humans over 30 years ago [2]. Then, it was found that bZIP-type TFs are involved in metabolism, development, cell cycle, reproduction, and programmed cell death [1]. In plants, the functions of bZIP-type TFs include abiotic stress response, metabolism, mediating pathogen defense, hormone signaling, and senescence [3]. bZIP-type TFs are involved in various biological processes in fungi (described below).

In *Saccharomyces cerevisiae*, the first bZIP-type TF called Yap (yeast activator protein) was found [4], then a subset of YAP TFs (YAP1-YAP8) have been well defined [5]. Several bZIP proteins have been reported in filamentous fungi, such as *Aspergillus* spp. [6,7,8,9,10,11], *Neurospora crassa* [12], plant pathogens *Fusarium graminearum* [13], *Magnaporthe oryzae* [14], and V*erticillium dahlia* [15], fungal endophyte *Epichloë festucae* and *Pestalotiopsis fici* [16,17], human pathogen *Aspergillus fumigatus* [18], and insect pathogen *Metarhizium robertsii* [19]. Furthermore, bZIP proteins from oomycetes and the mushroom have been characterized [20,21]. bZIP-type TFs in fungi mainly function in mediating development, stress responses, sexuality, secondary metabolism, and pathogenicity. A novel Yap-like bZIP, RsmA (restorer of secondary metabolism A), was found using a multicopy-suppressor approach in *Aspergillus nidulans* [22]. Overexpression of *rsmA* partially restores sterigmatocystin in *A*. *nidulans ∆laeA* strain [22]. Then, the regulation mechanism reveals that RsmA regulation on secondary metabolite production links to the response to environmental stresses through *aflR* [23]. More recently, several orthologs of RsmA have been characterized in the filamentous ascomycetes, including RsmA in *A*. *fumigatus*, MoRsmA/MoFcr3 in *M*. *oryzae*, and PfZipA in *P*. *fici* [17,18,24,25]. The RsmA ortholog from *A*. *fumigatus* positively regulates the biosynthesis of gliotoxin and its precursor cyclo (L-Phe-L-Ser) [18]. PfZipA in the endophytic fungus *P*. *fici* has an impact on oxidative stress response and secondary metabolites production [17]. However, no distinguishable phenotypes were observed in MoRsmA in plant pathogen *M*. *oryzae* [24,25]. However, the ortholog of RsmA has not been functionally studied in *Aspergillus flavus* as the producer of carcinogenic and mutagenic aflatoxins.

The saprotrophic fungus *A*. *flavus* as an opportunistic pathogen causes disease of several important crops, such as peanuts, treenuts, corn, and cottonseed [26]. It is notorious because it produces aflatoxins (AFs) as secondary metabolites in the seed both before and after harvest [26]. Besides leading to contaminant of food and feed, AFs especially AFB_1_ are potent carcinogens, and *A*. *flavus* is the second leading pathogen of aspergillosis [27]. The primary disseminating and infection unit of *A*. *flavus* is asexual spores. When the growth conditions are unfavorable, *A*. *flavus* forms the resting structure sclerotia capable of long periods of dormancy. Then, sclerotia can produce the primary inoculum during the next infection cycle [28].

Understanding how the production of secondary metabolites, fungal development, and virulence are regulated is critical to control Aspergillus diseases and AFs production. Several genes have been identified, which regulate the virulence and production of AFs [26,27,28]. Among the regulators, several bZIP transcription factors mediate virulence and secondary metabolism [29]. In *A*. *flavus*, bZIP TF MeaB regulates secondary metabolism and virulence [30]. AtfB regulates aflatoxin production by binding to the promoters of seven aflatoxin biosynthesis genes that carry CREs [10]. A recent study revealed that the bZIP transcription factor *Afap1* positively regulates the production of aflatoxin B_1_ and oxidative stress response [31]. Interestingly, bZIP transcription factor Yap-1Leu558Trp substitution in *A*. *flavus* is confirmed as being responsible for voriconazole-resistant phenotypes [32]. When *A*. *flavus* was exposed to piperine, transcript levels of bZIP transcription factors *atfA*, *atfB* and *ap-1* were increased [33]. Genome analysis of *A*. *flavus* identified 22 bZIP-type TFs [34], however, the functions of most bZIP-type TFs are still not clear.

In this study, the ortholog gene of bZIP-type TF *rsmA*, termed AflRsmA, was deleted and functionally characterized in *A*. *flavus*. We showed that AflRsmA mediates the regulation of oxidative stress response, sclerotia production, virulence against hosts, and aflatoxins production of *A*. *flavus*.

## 2. Results

### 2.1. Identification and Phylogenetic Analysis of AflRsmA, a RsmA ortholog in A. flavus

Since *A*. *nidulans* RsmA was a suppressor of a secondary metabolism mutant (i.e., *laeA*) through the activation of gene expression by binding to Yap-like sites [22,23], its ortholog in *A*. *flavus* could also regulate secondary metabolism. By searching the *A*. *flavus* whole genome using the RsmA amino acid sequence of *A*. *nidulans* as a query, a gene AFLA_133560 was identified with the highest identity to RsmA (65%). NCBI data showed that the AFLA_133560 ORF, consisting of 660 bp with an intron (102 bp), encodes a putative bZIP protein of 185 amino acids (aa). However, compared with the reported RsmA homologs in fungi, the length of the protein encoded by the AFLA_133560 gene is shorter. To identify the encoding sequence of the RsmA ortholog gene accurately, the ORF and cDNA were both amplified and sequenced, and the sequence was then analyzed using the DNAMAN 7.0 software (Lynnon Biosoft, San Ramon, CA, USA ) and SoftBerry software (Softberry, Inc., Mount Kisco, NY, USA) [35]. We found that *rsmA* ortholog gene in *A*. *flavus* is from the start codon of AFLA_133570 to the stop codon of AFLA_133560, consists of 1070 bp with two introns (47 and 102 bp), and encodes a 305 aa protein, henceforth called AflRsmA (Figure 1A). Besides the most conserved bZIP domain in all reported RsmA homologs, the redox domain was the conserved domain of AflRsmA in comparison with *A*. *nidulans* RsmA, *A*. *fumigatus* RsmA, *M*. *oryzae* MoRsmA, and *F*. *graminearum* GzbZIP020 (Figure 1B). Based on the bZIP domain sequences, a phylogenetic analysis with functionally characterized RsmA indicated that AflRsmA is more closely related to the RsmA in *A*. *nidulans* and *A*. *fumigatus* (Figure 1C).

### 2.2. Expression of AflrsmA and Creation of AflrsmA Mutant

To identify the function of *AflrsmA* using reverse genetics methods, we firstly examined the transcription level of *AflrsmA*. The transcription level of *AflrsmA* gene was detected by RT-PCR analysis, when *A*. *flavus* was grown on PDA at 29 °C for 3 days (Figure 2A). Then we created an *AflrsmA* deletion strain (*ΔAflrsmA*) by the replacement of *AflrsmA* with *pyrG* gene and an overexpression (OE) strain (*AflrsmA^OE^*) using the constitutive promoter *gpdA*(p) from *A. nidulans* in wild type (WT) background via PEG-mediated protoplast transformation (Figure 2B). The selected mutants were verified by diagnostic PCR. Furthermore, the knockout strains were verified using southern blot (Figure 2C), and the transcription level of the overexpression strains verified by RT-PCR showed that in comparison with WT, the transcript level of *AflrsmA* in *AflrsmA^OE^* strain was higher (Figure 2D).

### 2.3. AflRsmA Responses to Oxidative Stress in A. flavus

To investigate the tolerance of *A*. *flavus* stress to oxidant stress, the correct mutant strains were inoculated on YGT media containing oxidant reagents. Without oxidative stress, there was no difference in the colony diameters of WT, *ΔAflrsmA* and *AflrsmA^OE^* strains (Figure 3). Growth assays on YGT supplemented with various oxidative reagents demonstrated that *ΔAflrsmA* strain was less sensitive to tBOOH, but *AflRsmA*^OE^ strain was more sensitive to MSB, as determined by the diameter measurement (Figure 3). However, there is no difference in colony diameter among WT, *ΔAflrsmA* and *AflrsmA^OE^* strains with H_2_O_2_ treatment (Figure 3). Taken together, AflRsmA is critical for the response of *A*. *flavus* to oxidative stress, but the regulation mechanism is dependent on the oxidant reagent.

### 2.4. AflRsmA Involved in AFB_1_ Production is Oxidative Stress Related

To investigate the effect of AflRsmA on the biosynthesis of secondary metabolite AFB_1_, we assessed the AFB_1_ production on the artificial media YES (Yeast Extract Sucrose agar) and YGT (Yeast Extract Glucose Trace element agar). We found that the AFB_1_ production showed no difference between WT and *ΔAflrsmA*, however, there was increased AFB_1_ production from the *AflrsmA*^OE^ strain in comparison with WT on the artificial medium YES (Figure 4).

It is reported that secondary metabolite synthesis is linked with the oxidative stress. AFB_1_ production on YGT supplemented with oxidant stressor was also measured. When the strains were grown on YGT medium supplemented with MSB, or tBOOH, AFB_1_ production produced by *ΔAflrsmA* decreased compared to WT and *AflrsmA^OE^* (Figure 4). However, no obvious differences were observed in AFB_1_ production among WT, *ΔAflrsmA* and *AflrsmA^OE^* under H_2_O_2_-induced stress by TLC analysis (Figure 4). These results support that the role of AflRsmA in mediating AFB_1_ responses to oxidative stress with differential responses is dependent on which oxidative pathway is activated.

### 2.5. AflRsmA Positively Regulates Sclerotia Formation

Sclerotia are resistant structures of fungi which allow survival under adverse environmental conditions [37]. *A*. *flavus* is a kind of fungus that forms sclerotia capable of surviving environmental extremes. To study the role of AflRsmA on sclerotia formation, we examined sclerotial production in WT, *ΔAflrsmA* and *AflrsmA^OE^* strains grown on a modified Wickerham medium. The number of sclerotia of *ΔAflrsmA* strain was significantly decreased compared to that of WT strain (Figure 5). Conversely, the sclerotia formed by *AflrsmA^OE^* strain was significantly more than that by WT strain (Figure 5). The opposite effects of the deletion and overexpression of *AflrsmA* on sclerotia formation indicated that AflRsmA positively regulates the sclerotia formation.

### 2.6. AflrsmA is a Positive Regulator of Virulence toward Corn

To confirm whether AflRsmA is a regulator of *A*. *flavus* virulence, the effect of deletion and overexpression of *AflrsmA* on the production of spores and toxin AFB_1_ on corn were examined. Spores from WT, *ΔAflrsmA* and *AflrsmA^OE^* strains were inoculated on corn kernels. We found that overexpression of *AflrsmA* resulted in a significant decrease in the number of spores compared to WT, but a significant increase of spores compared to *ΔAflRsmA* strain (Figure 6A,B). However, among the WT, deletion, and overexpression strains, the change trend of AFB_1_ production was different from that of spore production. The AFB_1_ yield of the *AflrsmA^OE^* strain was higher than that of the WT strain. Conversely, the *ΔAflrsmA* strain produced less AFB_1_ in comparison to WT (Figure 6C). Thus, the above results suggest that AflRsmA may be involved in sporulation and AFB_1_ synthesis on corn by different mechanisms.

## 3. Discussion

Our interest in exploring the impact of AflRsmA (RsmA ortholog) on AFs production in *A*. *flavus* arose from the reported function of RsmA in the model fungus *A*. *nidulans* [22,23]. In *A*. *nidulans*, the bZIP protein RsmA (restorer of secondary metabolism A) as a novel Yap-like bZIP was first identified in a *ΔlaeA* strain, which regulates not only the production of the anti-predation metabolite sterigmatocystin but also the biosynthesis of anthraquinone asperthecin [22,23]. The RsmA ortholog in *A*. *fumigatus* was found to positively regulate the production of gliotoxin [18]. Over 70 new bioactive secondary metabolites were isolated from plant endophyte *P*. *fici*, such as the first chlorinated pupukeanane metabolite chloropupukeananin and its precursors pestheic acid and iso-A82775C [38]. A RsmA ortholog PfZipA from *P*. *fici* negatively regulates the production of isosulochrin and iso-A82775C on rice medium, but positively regulates pestheic acid on rice medium and isulochrin and ficipyroneA on PDA medium [17]. Similar to the reported RsmA from other fungi, AflRsmA in *A*. *flavus* contributes to the regulation of AFB_1_ production. Though RsmA is conserved in the regulation of SM production, there are several differences among fungi. Firstly, *AflrsmA* in *A*. *flavus*, *rsmA* in *A*. *fumigatus* and *PfzifA* in *P*. *fici* are low expressed genes, however, RsmA in *A*. *nidulans* is not expressed [17,18,22,23]. In contrast to medium-dependent regulation of secondary metabolites by PfZifA in *P*. *fici* [17], regulation of AFB_1_ production by AflRsmA was independent of medium in *A*. *flavus*, and *ΔAflrsmA* strain produced a similar TLC profile as WT strain. Surprisingly, *ΔAflrsmA* strain produces less AFB_1_ and spores than WT on host corn. According to the result, we speculate that the biomass reduction might have mainly led to a decline in AFB_1_ production in *ΔAflrsmA* strain. Similar to more multiple *gli* pathway metabolites produced by *A*. *fumigatus OErsmA* strain [18], more AFB_1_ was detected in *AflrsmA^OE^* strain not only on artificial media YGT and YES, but also on host plant corn. However, *AflrsmA^OE^* strain produced less spores than WT, but more AFB_1_ than WT, suggesting that *AflrsmA* overexpression improved the AFB_1_ producing ability of *A*. *flavus*.

In *A*. *nidulans*, RsmA binds to two sites in the promoter of the transcription factor *aflR*, which is required for sterigmatocystin and aflatoxin biosynthesis, and activates its expression [23,39,40]. The transcription factor RsmA binds to two sites in the bidirectional promoter region of aflR/aflJ in *A*. *nidulans*, TTAGTAA (Y) and TGACACA (R) with one base variation (underlined letter) [23]. The *A. flavus* AflRsmA shares 65% identity with RsmA in *A*. *nidulans*, which suggests that AflRsmA may regulate AFB_1_ biosynthesis by impacting the expression of *aflR* gene. Further analysis showed that the *A*. *flavus* AflR shares only 37.78% identity with that of *A*. *nidulans,* and AflS is the ortholog gene of *A*. *nidulans* AflJ with 34.13% identity. However, only one binding site TGACCCA was found in the bidirectional promoter region of *aflR*/*aflS* in *A*. *flavus*. Based on the above analysis, we reasoned that AflRsmA may regulate AFB_1_ biosynthesis in a novel way.

bZIP transcription factors are associated with oxidative stress response in aflatoxigenic *Aspergillus* species, such as *Afap1* in *A*. *flavus* and *Apyap1* in *A*. *parasiticus* [31,41]. In this study, we found that *A*. *flavus* AflRsmA was involved in response to oxidative stress like that reported in *A*. *fumigatus* and *P*. *fici*, but unlike that reported in *A*. *nidulans* [17,18,22]. However, contrast to less sensitive to MSB in *A*. *fumigatus OErsmA* mutant [18], *A*. *flavus AflrsmA^OE^* strain was more sensitive to MSB (Figure 3). Interestingly, deletion of *PfzifA* in *P*. *fici* resulted in less sensitivity to MSB, but *A*. *flavus ΔAflrsmA* strain and *A*. *fumigatus ΔrsmA* strain have the same sensitivity to MSB as their respective WT strain [17,18]. In contrast to its response to MSB, *ΔAflrsmA* is less sensitive to tBOOH, which is similar to *P*. *fici* [17]. Therefore, oxidative stress sensing of *rsmA* in *A*. *flavus*, *A*. *fumigatus* and *P*. *fici* may indicate that RsmA plays important role in the oxidative stress response in fungi, but the specific regulatory mechanism is species-dependent. The differential role of AfRsmA involved in oxidative stress response may be related with the action mechanism of oxidant. MSB increases the intracellular superoxide levels, while H_2_O_2_ generates peroxide anions [42]. AflRsmA may not regulate the genes involved in the oxidative stress pathway, which is activated by superoxide and peroxide. Following the previously reported membrane lipid peroxidation by tBOOH [43], we considered that the main role of AflRsmA may be membrane lipid peroxidation. In this study, these observations suggested that AflRsmA may directly or indirectly regulate genes that are involved in the response to oxidative stress in *A. flavus,* but it remains to be determined how AflRsmA regulate oxidative stress response.

Various biological roles of secondary metabolites produced by fungi are known, including stress tolerance, fungivore resistance, and quorum sensing [40,44,45]. The biosynthesis of a series of secondary metabolites is induced by ROS in aspergilli [46]. Similarly, if ROS is applied in vitro, aflatoxin production can be stimulated [47], such as aflatoxin biosynthesis stimulated by H_2_O_2_ in *A*. *flavus* [31]. Previous studies have shown that several transcription factors have been involved in oxidative stress and secondary metabolism, including numerous bZIP-type TFs [29,48,49]. In aspergilli, Yap1 orthologs were reported as the coregulator of oxidative stress response with secondary metabolism, such as *napA* in *A*. *nidulans*, *Aoyap1* in *A*. *ochraceus*, *Apyap1* in *A*. *parasiticus*, and *Afap1* in *A*. *flavus* [6,9,10,31]. Among AP-1 reported in aspergillus, deletion of *ApyapA* of *A*. *parasiticus* resulted in a strain that was more sensitivity to extracellular oxidants and produced more aflatoxin, while the *Afap1* disruptant increased sensitivity to H_2_O_2_, but decreased AFB_1_ production in *A*. *flavus* [31,41]. Deletion of *Agyap1* in *Ashbya gossypii* resulted in a higher sensitivity to oxidative stress, but decreased riboflavin production [50]. In the plant pathogen *F*. *graminearum*, *Fgap1* is involved in the response to oxidative stress and trichothecene production [51]. Besides, the transcription factor AtfB, a member of the bZIP/CREB family, integrates oxidative stress with secondary metabolism in aspergillus, such as *A*. *flavus* [11]. A recent study showed that H_2_O_2_ promotes the production of AFB_1_ in *A. flavus* [31]. It was found that the AFB_1_ production was dependent on AflRsmA in this study. Furthermore, AFB_1_ production was altered by oxidative stress treatment in *A*. *flavus ΔAflrsmA* strain, which was completely different from that with no oxidative stress treatment. In general, the change trend of AFB_1_ production in the *ΔAflrsmA* strain under MSB-inducing stress is similar to that with tBOOH treatment (Figure 4). However, under oxidative stress induced by H_2_O_2_, the low sensitivity of TLC analysis may result in no obvious difference of AFB_1_ production being observed between WT and *ΔAflrsmA*. This suggests that AflRsmA as a regulator plays an important role in AFB_1_ production. In plant endophyte *P*. *fici*, when many compounds were detected, PfZipA was found to not only positively regulate several compounds, but also negatively regulate some compounds against oxidative stress [17]. In view of the above, we speculate that the regulation mechanism of AflRsmA is more complex. Further analysis of more secondary metabolites would reveal more detailed functions of AflRsmA in *A*. *flavus*. In future studies, we would address how AflRsmA regulates AFB_1_ production by oxidative stress response in artificial media, and determine what this regulatory pattern is in host.

The change of secondary metabolite production has been correlated with conidiophore formation and sclerotia production [28,52]. A number of genetic regulators were found to not only control the formation of developmental structures, such as spores and sclerotia, but also govern the secondary metabolites production in fungi [52,53]. In *A. flavus*, several genetic co-regulators were found, which activate the genes involved in secondary metabolites production and formation of spores and sclerotia. For example, more conidia, but no sclerotium, were produced in *A. flavus ΔveA* strain. Importantly, *veA* was required for the production of AFs, CPA, and asparasone in *A. flavus*, which have been isolated from the *A. flavus* sclerotia [28,54,55]. Compared to WT, overexpression of *AflrsmA* increases sclerotia and AFB_1_ production, while the deletion of *AflrsmA* gene decreases sclerotia but has no effect on AFB_1_ production. These suggest that *AflrsmA* may be a co-regulator of secondary metabolism and sclerotia production. However, detection of other secondary metabolites, such as CPA and asparasone involved in sclerotia formation has not been performed. Thus, we should determine whether the production of CPA and asparasone is regulated by AflRsmA in the future.

In conclusion, the bZIP transcription factor AflRsmA were characterized in the an aflatoxigenic *A*. *flavus*. The AflrsmA gene involves in the oxidative stress response dependent on specific oxidant and sclerotia formation, and regulates the AFB_1_ biosynthesis by the oxidative stress response pathways. The results will provide comprehensive understanding as to how *A*. *flavus* responses the host defense.

## 4. Materials and Methods

### 4.1. Strains, Media and Culture Conditions

*A. flavus* strains and plasmid used and generated in this study were listed in Table 1. All strains were maintained as glycerol stocks at −80 °C. All strains were grown on potato dextrose agar (PDA, Difco) for spore production at 37 °C for five days in the dark or for RNA extraction at 29 °C for 48 h in the dark. Liquid YGT medium was used to collect the mycelia for DNA extraction, and YGT agar plates were used to study oxidative stress susceptibility. For sclerotia production, spores were maintained on a Wickerham medium at 37 °C for 7 days in the dark. YGT media were supplemented with 10 mM uridine and 10 mM uracil as needed.

### 4.2. Identification, Phylogentic Analysis and Domain Prediction of AflrsmA

We used the protein sequence of RsmA from *A*. *nidulans* (EAA60905.1) as the query to perform BLAST search to find its homologous genes in *A*. *flavus*. To predict the protein domain, the proteins were performed manually using InterProScan 5 on EBI web server [36], including FCR3 from *Candida albicans* (accession no. Q8X229), Yap3 from *S*. *cerevisiae* (accession no. NP_011854.1), RsmA from *A*. *fumigatus* (accession no. XP_749389), RsmA from *A*. *nidulans* (accession no. XP_662166), MoRsmA from *M*. *oryzae* (accession no. XP_003721157.1), and GzbZIP020 from *F*. *graminearum* (accession no. ESU14603). Then, the domains were visualized using IBS software [57]. The protein sequences of bZIP-type TFs RsmA from fungi functionally verified by the experiment were aligned with the homologue gene from *A*. *flavus* with Clustal X 2.0. A maximum-likelihood tree was generated using a JTT Matrix model with 1000 replicates for bootstrapping using the program MEGA 7.0 [58].

### 4.3. RNA Extraction of A. flavus and Reverse Transcription-Polymerase Chain Reaction (RT-PCR) of AflrsmA

*A. flavus* was cultured at 29 °C on PDA media for 48 h. Then, mycelia and spores were harvested and total RNA was extracted using TRIzol_Reagent (ambion) (Life Technologies, Carlsbad, CA, USA). DNA was digested with RNase-free DNase I (ThermoFisher Scientific, Waltham, MA, USA). Subsequently, RNA was reverse transcribed into first-strand cDNA using the RevertAid First Strand cDNA Synthesis Kit (Thermo Fisher Scientific, Waltham, MA, USA). To assess the transcription level of *AflrsmA*, the sequence of this gene coding region was amplified using primer pair 133560RT_F and 133560RT_R (Table 2). The internal control was the house-keeping gene *actin*.

To identify *AflRsmA* accurately, the *AflrsmA* coding gene was sequenced. The fragments were amplified from *A*. *flavus* genomic DNA and cDNA using primer the pair 133560L/F and 133560/R and the pair 133560/F and 133560/R, respectively. Then, the PCR product was sequenced at Biosune biotechnology (Shanghai, China) Co., Ltd. China. MEGA 7.0 was used to obtain consensus sequences from DNA sequences generated from forward and reverse primers. Then, the consensus sequence was compared with DNA and cDNA sequences from NCBI’s GenBank database using the software DNAMAN 7.0. The protein sequence was predicted using the software SoftBerry [35].

### 4.4. Creation of AflrsmA Mutant Strains

The PCR primers sequences are listed in Table 2. TransStart_FastPfu DNA polymerase (Transgene Biotech) was used for PCR amplification. For the creation of *AflrsmA* deletion strains of *A*. *flavus* (*ΔAflrsmA*), 1.1 kb upstream and 1.1 kb downstream fragments of the part of *AflrsmA* AFLA_133560 were amplified from *A*. *flavus* genomic DNA, respectively. Meanwhile, *A*. *fumigatus* orotidine-5′-monophosphate decarboxylase (*pyrG*) gene was amplified. Then, a 3.8 kb deletion cassette of *AflrsmA* carrying *AflrsmA* bZIP domain upstream fragment, *A*. *fumigatus pyrG*, and *AflrsmA* downstream fragment was constructed by fusion PCR, then transformed into *A*. *flavus* recipient strain [59]. To construct *AflRsmA* overexpression (*AflRsmA^OE^*) strains, *AflrsmA* coding region with 0.28 kb terminator was amplified from *A*. *flavus* genomic DNA using primers RsmA/F and RsmA/R. Meanwhile, the *gpdA* promoter and *A*. *fumigatus pyrG* were amplified from the plasmid pWY25.16 and the *A*. *fumigatus*, respectively. Then, the fragments were combined using fusion PCR [59], then the overexpression cassette was transformed into *A*. *flavus*. *A*. *flavus* protoplast preparation and fungal transformation were both performed as the described method [53]. The disruption and overexpression mutants were firstly verified using diagnostic PCR with primers inside and outside the corresponding gene (Table 2). Then, disruption strains were further verified by southern blot, and overexpression strains were further verified by RT-PCR analysis.

### 4.5. Oxidative Stress Susceptibility of A. flavus Mutants

The different concentrations of oxidation reagents were used to estimate the sensitivity of the mutants to oxidative stress, including the following agents (concentrations and mechanisms of action were given in parentheses): tert-Butyl hydroperoxide (tBOOH; 1.2 mM, 1.4 mM, and 1.6 mM; accelerates lipid peroxidation), menadione sodium bisulfite (MSB; 0.1 mM, 0.2 mM, and 0.4 mM; increases intracellular superoxide concentrations), and H_2_O_2_ (6 mM, 8 mM, and 10 mM; increases intracellular peroxide concentrations). The indicators of condition optimization are the colony diameters. Two hundred freshly grown (5 days) conidia were suspended in 2 μL 0.1% Tween 80 and were spotted on YGT plates supplemented with the optimal oxidation agents. All stress plates were incubated at 29 °C for 5 days. During the growth process, the colony diameters we measured from the third day to the fifth day. The appropriate concentrations of tBOOH (1.2 mM) and MSB (0.2 mM), H_2_O_2_ (6 mM) were chosen and used in the stress sensitivity assays. Three replicates were included in all treatments, and the whole experiment was repeated three times.

### 4.6. Analysis of Aflatoxin B_1_ Produced by A. flavus

Approximately 10^5^ spores of WT and *AflRsmA* mutants were spotted onto a Petri dish plate (60 × 60 mm) containing 10 mL YES medium (15% sucrose, 2% yeast extract, and 1% soytone, pH 5.5), and incubated at 29 °C for five days. For the YGT medium, the same cultures used for oxidative stress assessment were extracted for aflatoxin B_1_ assessment using thin layer chromatography (TLC) on the fifth day. Briefly, seven agar plugs (7 mm diameter/each plug) were taken from each plate, then transferred to a 4 mL Eppendorf tube. The plugs were extracted with 2 mL dichloroethane in each tube by sonication for 1 h at room temperature. The extracts were dried completely at room temperature, then suspended in 100 μL dichloroethane. In total, 5 μL extract and AFB_1_ standard were applied to the TLC plates respectively. TLC plates were developed using dichloroethane: acetone (90:10, vol/vol) solvent system and visualized under 254 nm light. All treatments included three biological replicates, and the whole experiment were repeated three times.

### 4.7. Assay for Sclerotia Production

For quantitative analysis of sclerotial production, 10^4^ freshly grown (5 days) spores were spotted on modified Wickerham medium plates, which were suspended in 2 μL 0.1% Tween 80. The plates were incubated at 37 °C for 7 days in dark conditions. 70% ethanol was sprayed on the plates to kill and wash away conidia to help the enumeration of sclerotia. Then, seven agar plugs (7 mm diameter/plug) were taken from the diameter of each plate and the number of sclerotia was counted. Three replicates were included in all treatments, and the whole experiment was repeated three times.

### 4.8. Corn Infection and Aflatoxin Extraction from Seed

The corn infection experiment was performed as described by Chang et al. [60]. Prior to inoculation, the seeds were surface-sterilized. Firstly, the seeds were immersed in 70% ethanol that contained 0.02% Triton-X 100 with gentle shaking for 4 min to surface-sterilize the seeds; then, the seeds were rinsed three times with sterile water containing 0.02% Triton-X 100. Eight seeds as a group were placed onto a Petri dish plate (60 × 60 mm), briefly air dried, and each seed was inoculated with 2 μL of a freshly grown conidial suspension (two thousand conidial suspended in 0.01% Tween 80 solution). The control was seeds that were each spotted with 2 μL 0.01% Tween 80 solution. During the whole infection process, five hundred milliliters of sterile water was added to every Petri dish plate each day. The plates were incubated at 29 °C in the dark for five days. After incubation, pictures of the plates were taken. Eight seeds from each plate were transferred to 50 mL tubes containing 5 mL of the 0.01% Tween 80 solution, then the tube was vortexed vigorously for 2 min to dislodge conidia. In total, 1 mL suspension from each tube was used to count conidia using hemocytometer, and the rest of the suspension was extracted with 4 mL dichloroethane. The extracts were dried completely at room temperature, then were resuspended in 100 μL dichloroethane. Five μL extract and AFB_1_ standard was applied to the TLC plates respectively. The TLC plates were developed using dichloroethane:acetone (90:10, vol/vol) solvent system and visualized under 254 nm light. All treatments included three biological replicates and the whole experiment was repeated three times.

### 4.9. Statistical Analysis

For statistical analysis, GraphPad Instat software package, version 5.01 (GraphPad software Inc.) were used to analyze data, according to the Tukey–Kramer multiple comparison test at *p* < 0.05. The different letters indicate the significant difference between data.

## Figures and Tables

**Figure 1 toxins-12-00271-f001:**
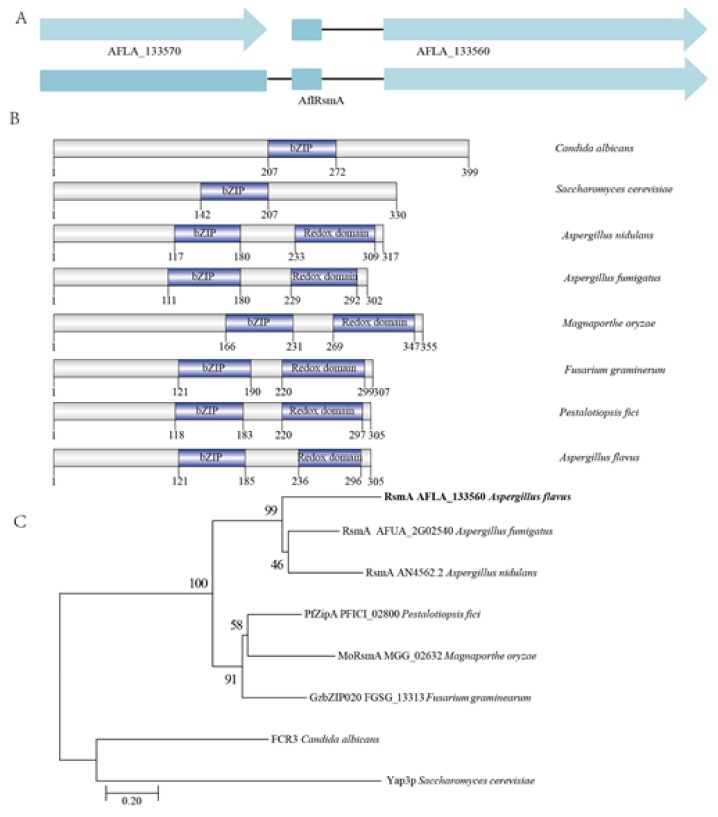
Gene structure, functional domain, and phylogenetic analysis of *A*. *flavus* AflRsmA. (**A**) Gene structure of *A*. *flavus AflrsmA*. The upper lane indicates gene structure of *AflrsmA* (AFLA_133560) and the adjacent gene (AFLA_133570) from NCBI database. The lower lane indicates gene structure of *AflrsmA* identified in this study. The blue box and the line represent the exons and the introns of *AflrsmA* gene, respectively. (**B**) Functional domains of *A*. *flavus* bZIP-type TF AflRsmA. bZIP, basic leucine zipper domain; Redox domain, Yap1 redox domain. The number under the protein indicates the position of domain and the protein length. The protein domains were predicted manually using InterProScan 5 on EBI web server [36]. *Candida albicans* FCR3 (Q8X229), *S. cerevisiae* Yap3 (NP_011854.1); *A. nidulans* RsmA (AN4562.2, XP_662166), *A*. *fumigatus* RsmA (AFUA_2G02540, XP_749389), *M*. *oryzae* MoRsmA (MGG_02632, XP_003721157.1), and *F*. *graminearum* GzbZIP020 (FGSG_13313, ESU14603). (**C**). Phylogenetic analysis of bZIP-type TF RsmA from and RsmA orthologs that have been functionally verified in different fungi. The protein sequences were aligned with Clustal X and the maximum likelihood tree was generated using MEGA7.0 software. RsmA from *A*. *flavus* is in bold.

**Figure 2 toxins-12-00271-f002:**
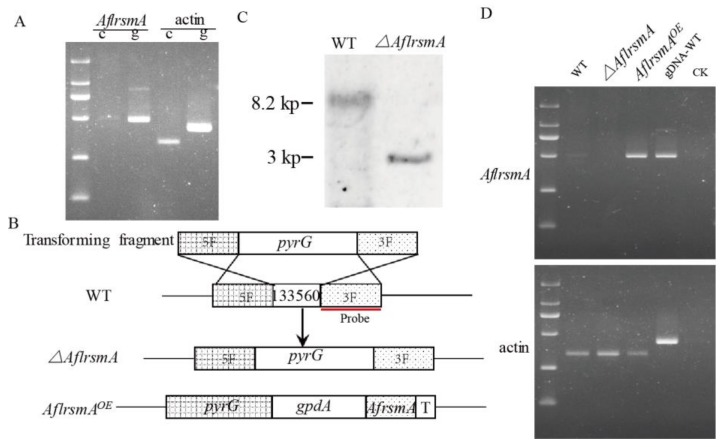
Generation of *AflrsmA* deletion and overexpression strains. (**A**). Transcription level analysis of *AflrsmA* by RT-PCR. (**B**). Schematic illustration of the deletion and overexpression of *AflrsmA* gene. The selection marker is the *pyrG* gene from *A*. *fumigatus*. (**C**). *AflrsmA* deletion strain verified by southern blot analysis. PCR fragment of 3′ flanking region was used as the probe. Genomic DNA from WT and *ΔAflrsmA* strains were digested with *XhoI*. The expected size is 8.2 kb and 3 kp for WT and for *ΔAflrsmA* respectively. (**D**). Transcription level of *AflrsmA* gene in overexpression strain verified by RT-PCR. The internal control gene is actin gene.

**Figure 3 toxins-12-00271-f003:**
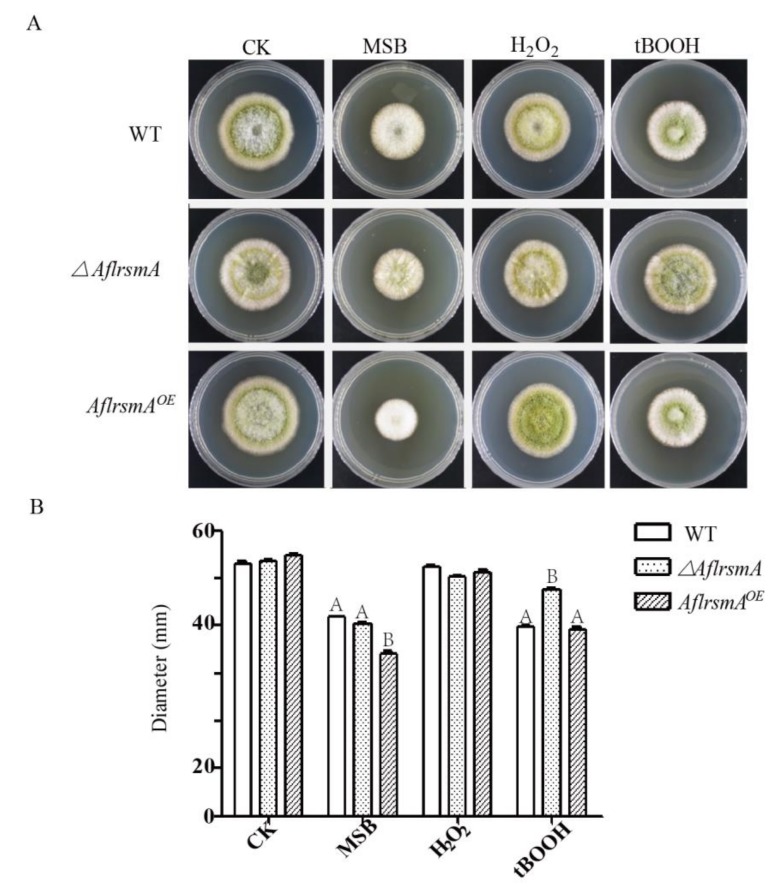
Comparison of the oxidative stress tolerance of *A*. *flavus* WT strains and mutant. (**A**) Mycelia growth of the *A*. *flavus* WT and AflrsmA mutants under oxidative stress. Two hundred conidia of *A*. *flavus* WT and *AflrsmA* mutants were cultured on YGT media supplemented with or without MSB (0.2 mM), H_2_O_2_ (6 mM), or tBOOH (1.2 mM) at 29 °C for five days. (**B**) Statistical analysis of colony diameter of the testing strains measured on the 5th day. Each treatment included three replicates. Error bars represent the standard deviations.

**Figure 4 toxins-12-00271-f004:**
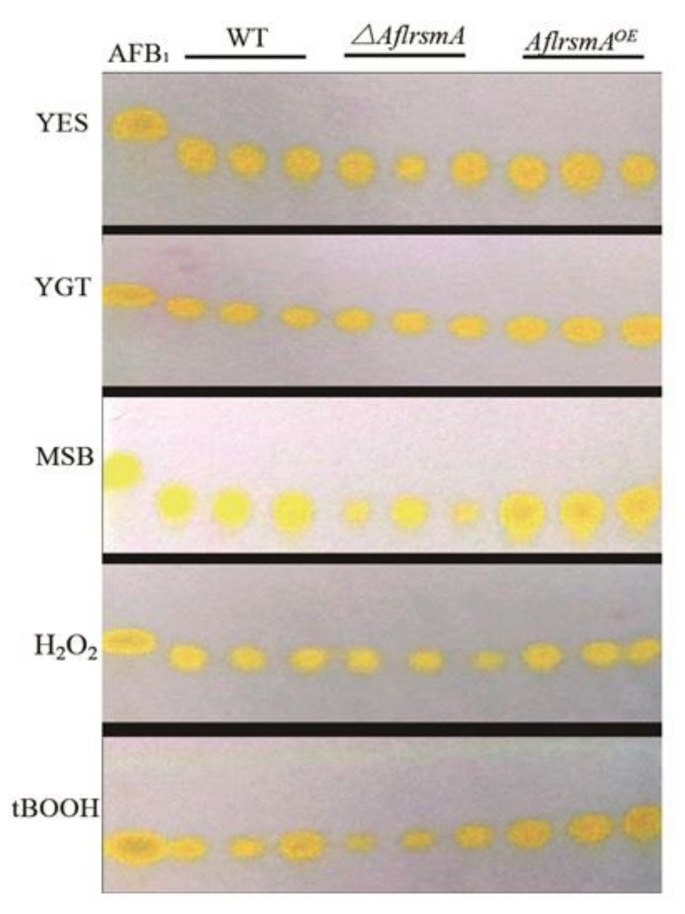
Assessment of Aflatoxin B_1_ production of *A*. *flavus* WT and AflrsmA mutants under various conditions by thin-layer chromatography. All testing strains were cultured on YES medium, or YGT medium with or without MSB (0.2 mM), H_2_O_2_ (6 mM) and tBOOH (1.2 mM) for five days at 29 °C. AFB_1_ = aflatoxin B_1_ standard.

**Figure 5 toxins-12-00271-f005:**
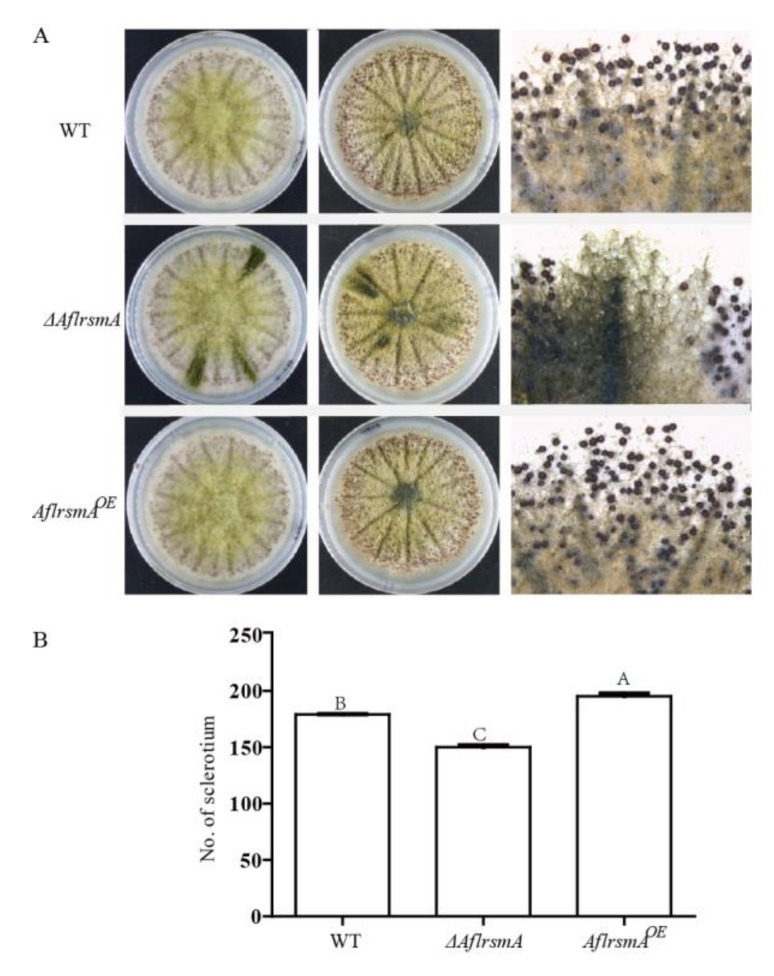
Sclerotial formation by *A*. *flavus* WT and *AflrsmA* mutants. (**A**) Visual phenotype of sclerotia production. 10^3^ conidia/plate was incubation on Wickerham medium and cultured for seven days at 37 °C. (**B**) Variation in sclerotial production by the WT and *AflsrmA* mutants on the Wickerham medium for seven days. The letter represents a significant difference at the level *p* < 0.05. Errors bars represent standard errors from three replicates.

**Figure 6 toxins-12-00271-f006:**
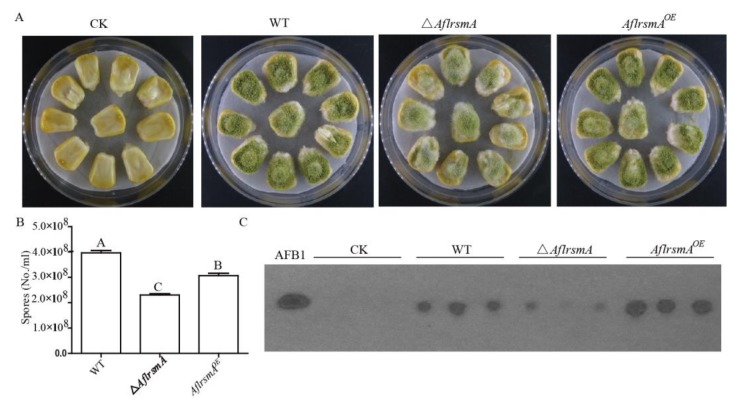
Effect of *AflrsmA* on *A*. *flavus* Pathogenicity. (**A**) Growth of fungal colonies on living corn after five days of inoculation. (**B**) Spore production on corn after five days of inoculation. Data were analyzed using the GraphPad Instat software package version 5.01 (Graph Pad software Inc., San Diego, CA, USA) according to the Tukey–Kramer multiple comparison test at *p* < 0.05. The letter indicates statistical significance at *p* < 0.05. (**C**) Thin layer chromatography analysis of aflatoxin B_1_ extracted from host plant corn. AFB_1_ = aflatoxin B_1_ standard.

**Table 1 toxins-12-00271-t001:** Fungal strains and plasmid used in this study.

Strain/Plasmid	Description	Reference
Recipient strain	PTS∆ku70∆pyrG	[56]
Wild type	*PTS∆ku70∆pyrG: AfpyrG*	[56]
*∆AflrsmA*	*pyg: ∆AflrsmA: A. flavus ∆ku70*	This study
*AflrsmA^OE^*	*A. fumigatus pyg: gpdA(p):AflrsmA: A. flavus ∆ku70*	This study
pWY25.16	*A. fumigatus pyroA:gpdA* in pGEMT easy vector	[23]

pXX, plasmid.

**Table 2 toxins-12-00271-t002:** PCR primers used in this study.

Name	Oligonucleotides Sequence (5’-3’)	Uses
133560RT_F	CACAGAACAGAGCAGCGTAG	For RT-PCR
133560RT_R	CTCATGCCCTTGGATTAGG	For RT-PCR
actin_F	CAGCCGCTAAGAGTTCCAG	For RT-PCR
actin_R	CACCGATCCAAACCGAGTAC	For RT-PCR
133560/5FE_F	GCGTCATGCGAGATTGTTTCC	5′ flanks amplification and for identification of 83100 mutant
133560/5F_R	GGGTGAAGAGCATTGTTTGAGGCCACTTCCAGAGGCGGAATGCTTC	5′ flanks amplification
133560/3F_F	GCATCAGTGCCTCCTCTCAGACGGCGTATTCGGTTCACGGTCATAATG	3′ flanks amplification
133560/3FE_R	GCTACTGGGTCTCAGAGTTGCTTC	3′ flanks amplification and for identification of 83100 mutant
133560/5F_F	GACGCTGGACACCTGAAACCCAAC	For whole length of KO cassette
133560/3F_R	CCTACTGTACCGATAACATGC	For whole length of KO cassette
*pyrG*_TF	GCCAGTACGAGTGTTGTGGAG	For identification of 133560 mutant
*pyrG*_TR	GTCAGACACAGAATAACTCTC	For identification of 133560 mutant
pyr*G*/F	GCCTCAAACAATGCTCTTCACCC	*pyrG* amplification
*pyrG*/R	GTCTGAGAGGAGGCACTGATGC	*pyrG* amplification
*gpdA*/F	GCATCAGTGCCTCCTCTCAGACCATCCGGATGTCGAAGGCTTG	*gpdA* amplification
*gpdA*/R	GTGTGATGTCTGCTCAAGCG	*gpdA* amplification
RsmA/F	GCTACCCCGCTTGAGCAGACATCACACATGACTCCAGCGAATCG	133560 ORF amplification
RsmA/R	GAGTTTGAGGTGCAGCTGG	133560 ORF amplification
133560/F	ATGACTCCAGCGCAATCG	For *AflrsmA* ORF identification
133560L/F	ATGGAGTACCCATACTATCC	For *AflrsmA* ORF identification
133560/R	TCAGAGAAGGTCATCATCATTAG	For *AflrsmA* ORF identification

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
