# Peer review of "The bZIP Transcription Factor AflRsmA Regulates Aflatoxin B_1_ Biosynthesis, Oxidative Stress Response and Sclerotium Formation in *Aspergillus flavus"

_toxins, 2020, doi:10.3390/toxins12040271_

Round 1

Reviewer 1 Report

general remarks:

In the manuscript at hand, the authors describe the involvement of the co-factor RsmA on aflatoxin B1 production and sclerotium formarion by Aspergillus flavus. Special emphasis is given to the influence of oxidative stress.

The experimental design is sound and was largely carried out in sufficient quality, the research is original and of a certain relevance to the field. The manuscript requires some improvements, though. Particularly, results and discussion chapters should be somewhat reorganized.

Thus, I advise publishing of the manuscript after a thorough revision of the following issues:

major concerns:

  • Results should be more clearly separated from discussion. At the moment, results are not only presented, but partially also interpreted in the results section. I advise a careful reorganization of this section, and moving respective discussions were they belong. Also, literature references and corresponding statements should be included in the discussion, not the results chapter.
  • figure 2: Here, the authors refer to “gene expression” and “overexpression”, however they show results at the RNA level, which per definition is the level of gene transcription, not expression. The latter would be visible only at the protein level. Authors should at least revise the text accordingly. If possible, a confirmation of results at the protein level should be pursued.
  • figure 4: Did you try to do a semi-quantitative analysis with subsequent statistical analysis? To me it looks like the ΔAflrsmA strain does produce less AFB1 with H2O2 treatment as compared to AflsrmAOE, despite being reported otherwise in line 174.
    If such an analysis is not possible, respective discussion of this result should be carefully reconsidered and relativized.
  • At the end of the discussion, there should be 1-2 sentences of conclusion, summarizing the key results and their relevance to the field.

minor suggestions:

  • AFB1 is usually abbreviated with subscript: AFB1
  • English can be improved here and there.
  • line 6: abbreviating “secondary metabolite” is quite unusual. It’s surely a matter of taste, I would prefer sticking to the whole phrase throughout the manuscript for the sake of reader’s intelligibility.
  • line 64/65: rephrase “produces secondary metabolites aflatoxins”, maybe to “produces aflatoxins as secondary metabolites”?
  • line 94, 99: please specify “bioinformatics analysis/method”
  • line 222-223: If you want to invent compounds like pestheic acid, isulochrin and ficipyroneA, you should explain their relevance. You should not take it for granted that readers know such details.
  • line 225: “similar to […] fungi,…” A.flavus is a fungi too. Rephrase.
  • line 248: “As we all know” I did not know before reading your manuscript. ;-) Consider rephrasing.
  • line 278: “Recent study shows…” à “A recent study…”; also “increases with increased” Consider rephrasing.
  • line 283: “a regulator of AFB1 and oxidative stress”: Regulator of AFB1 production, yes. But regulator of oxidative stress? Consider rephrasing.

Reviewer 2 Report

This is a review of “The bZIP Transcription Factor AflRsmA Regulates 3 Aflatoxin B1 Biosynthesis, Oxidative Stress Response 4 and Sclerotium Formation in Aspergillus flavus.”In this manuscript the authors describe the phenotypes of a A. flavus AflRsmA mutant that is either deleted for an ortholog of the A. nidulans gene or which over-expresses the transcription factor. They measure sensitivity to different oxidizing agents and correlate that sensitivity to aflatoxin and spore production. Overall while they find little difference in phenotype between the wild type and the mutant, but they do find a difference in phenotype between the wild type and the over-expressing strain, based on growth on oxidizing agents and overproduction of aflatoxin. They conclude that the AflRsmA transcription factor controls responses to oxidizing stress and production of aflatoxin. Overall, the data appear solid; however. the response to all oxidizing agents is not the same. This reviewer had several points to make that could add to the clarity of the manuscript

However, some of the data can be clarified and the text could be modified.

Major pts:

  1. The authors could better explain why over-expressing a transcription factor would actually enhance oxidative stress. Which genes may actually be upregulated?
  2. The data in some figures are clearer than others. For example, Figure 6 is very clear, whereas Figure 4 is not. The problem is that the blue on gray background in Figure 4 does not yield good contrast. My recommendation would be to have some quantitative number associated with the signal, such as what could be obtained by densitometer.
  3. The number of colonies is not explicit in several figures, such as Figure 3. The number of colonies should be detailed in the figure caption or materials and methods.
  4. The difference in phenotype concerning the resistance phenotype with hydrogen peroxide and the other oxidizing agents is interesting. The authors do mention that it might have something to do with membrane damage and supply a reference. This reviewer thinks that this point can be elaborated.

Minor points:

  1. The authors could trim some of the wording. It is not necessary to use “It is generally believed that….” This reviewer feels it is more appropriate to just state the observation and the reference, when applicable.

Reviewer 3 Report

Congratulations to your very high quality (eye opener introduction, well thought-out text, correct citations etc.) MS with relevant results. Please find my some comments.

Not acceptable methodology (and its statements) for the reviewer:

The patches of thin layer chromatography are not quantified (See fig 4. & 6.).

ad 158: A. flavus (italic)

ad 276: 43→46

Grammatic errors:

ad 160, 161, 258, 459
